# A Techno-Economic Analysis of Vehicle-to-Building: Battery Degradation and Efficiency Analysis in the Context of Coordinated Electric Vehicle Charging

**Stefan Englberger \*** , **Holger Hesse** , **Daniel Kucevic** and **Andreas Jossen**

Institute for Electrical Energy Storage Technology, Technical University of Munich (TUM),
Arcisstr. 21, 80333 Munich, Germany; holger.hesse@tum.de (H.H.); daniel.kucevic@tum.de (D.K.);
andreas.jossen@tum.de (A.J.)
\* Correspondence: stefan.englberger@tum.de; Tel.: +49-89-289-26969

**Abstract:** In the context of the increased acceptance and usage of electric vehicles (EVs), vehicle-to-building (V2B) has proven to be a new and promising use case. Although this topic is already being discussed in literature, there is still a lack of experience on how such a system, of allowing bidirectional power flows between an EV and building, will work in a residential environment. The challenge is to optimize the interplay of electrical load, photovoltaic (PV) generation, EV, and optionally a home energy storage system (HES). In total, fourteen different scenarios are explored for a German household. A two-step approach is used, which combines a computationally efficient linear optimizer with a detailed modelling of the non-linear effects on the battery. The change in battery degradation, storage system efficiency, and operating expenses (OPEX) as a result of different, unidirectional and bidirectional, EV charging schemes is examined for both an EV battery and a HES. The simulations show that optimizing unidirectional charging can improve the OPEX by 15%. The addition of V2B leads to a further 11% cost reduction, however, this corresponds with a 12% decrease in EV battery lifetime. Techno-economic analysis reveals that the V2B charging solution with no HES leads to strong self-consumption improvements (EUR 1381 savings over ten years), whereas, this charging scheme would not be justified for a residential prosumer with a HES (only EUR 160 savings).

**Keywords:** battery degradation; battery energy storage system; charging scheme; efficiency; electric vehicle; linear programming; lithium ion battery; operating expenses; residential battery storage; vehicle-to-building

## 1. Introduction

Increasing environmental awareness, technical improvements, and favorable regulatory conditions have all allowed the market for electric vehicles (EVs) in Germany and worldwide to experience an upturn [1,2]. Simultaneously, an increasing number of electricity consumers are investing in renewable energy sources. Photovoltaic (PV) power generators especially benefit from a growing popularity in residential homes, allowing these customers to reduce electricity costs and rendering them as prosumers [3]. A home energy storage system (HES) can be added to further increase self-consumption and self-sufficiency rates [4].

In literature, HESs and EVs are well-researched topics [4–6], however, combined approaches of both storage systems are still a very young research field [7]. While recent literature presents a novel energy management system (EMS) for residential buildings with HES and EV, the contribution comes short on analyzing the technical characteristics of the battery energy storage systems (BESSs) at varying

charging schemes [7]. In this work, we analyze how the aforementioned trends may interact, conduct a full techno-economic system analysis and reveal how prosumers with an EV may be able to optimize their electricity expenses. In particular, the degradation and efficiency of the HES and the EV's BESS are discussed. In addition, operating expenses (OPEX) are analyzed in the context of electricity costs for both the building and the vehicle. To increase the comparability of the results, a vehicle with an internal combustion engine (ICE) serves as a reference case.

As illustrated in Figure 1, three different charging strategies for the EV are analyzed and compared: Simple charging (SC) and optimized charging (OC) schemes, which both allow unidirectional power flows from the building to the vehicle, and the vehicle-to-building (V2B) strategy, which is an extension of the OC scheme allowing bidirectional power flows [6,8]. It is known that vehicle usage patterns may vary strongly [9]. For this reason, to make more valid statements about the degradation behavior, efficiency, and OPEX, the vehicle utilization patterns of a commuter and a supplementary vehicle are investigated. These vehicles are characterized by varying plug-in times at the power outlet of the prosumer's residence. As an additional degree of freedom, interaction between the EV battery and an optional stationary HES is examined. Particularly, the influence on the degradation and the efficiency of such a scenario considering two BESSs (EV and HES) is discussed. For the sake of simplicity, throughout this work, a typical German household with corresponding load and PV generator profiles is utilized and price signals of the German energy market are incorporated. However, the methodology can be applied to other profile data and the conclusive results drawn in this contribution are valid for other regions worldwide. An overview of the discussed simulation structure is visualized in Figure 1.

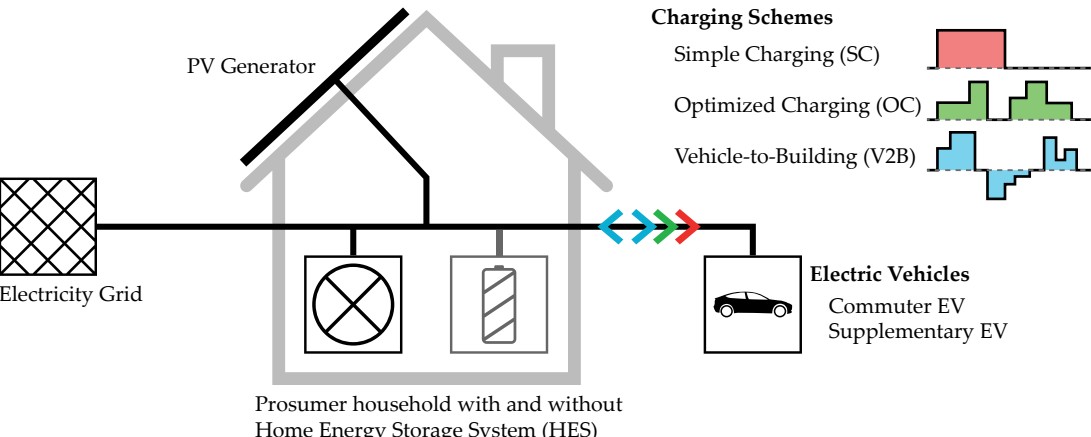

**Figure 1.** Schematic structure of the simulation environment of a prosumer household with three varying simulation dimensions: Consideration of home energy storage system (HES), two electric vehicle (EV) utilization patterns (commuter and supplementary car), and three different charging schemes (SC, OC, V2B).

The investigated scenarios in this work are simulated using a two-step approach. First, the residential power flow (RPF) model with an underlying linear programming (LP) algorithm optimizes the power flows within the residential multi-node system. Next, the optimized power flows are transferred to the open source simulation tool *SimSES* in order to model the resulting battery degradation and system efficiency [10].

This paper is structured as follows: Section 2 explains the optimization and simulation models as well as the system's topology, Section 3 presents the simulation results, and Section 4 concludes with a summary and discussion.

## 2. Methods

In order to optimize the electricity exchange between components and analyze the storage systems in a detailed fashion, two solution methods are combined, as is illustrated in Figure 2.

First, the power flows between the individual technical units are optimized using the RPF model. The underlying algorithm is based on LP, derived from the MATLAB optimization toolbox and the Gurobi optimizer [11]. Then, the simulation tool *SimSES* is used, which is capable of simulating the technical parameters of an energy storage system [10]. The results of the linear optimization are transferred to *SimSES* and represent the inputted alternating current (AC) power values of the energy storage system's inverter. By using *SimSES*' integrated operation strategy *PowerFollow*, the predefined time-discrete power values are implemented, and a detailed simulation is carried out. Both tools, the RPF model and the *SimSES* simulations are conducted in MathWorks MATLAB R2018b, operating at a sampling rate of 15 min [5].

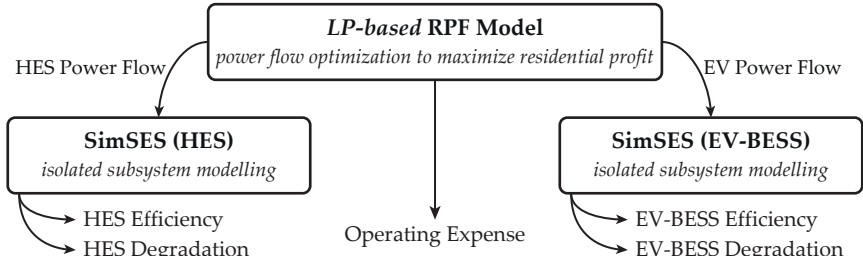

**Figure 2.** Schematic diagram of the two-step model structure, consisting of a linear programming (LP) based residential power flow (RPF) model, which optimizes the power flows so that the operating expenses (OPEX) are minimized, and the simulation tool *SimSES*, which validates the technical characteristics, round-trip efficiency, and battery degradation of the battery energy storage systems (BESSs).

The profit of a residential electricity prosumer in Germany is computed by simulating several different system configurations: Optional HES, optional EV, three different EV charging schemes, and two vehicle usage patterns.

Depending on the scenario, the RPF model of the investigated household consists of up to six main components, which are illustrated in Figure 3. The household is equipped with a PV generator with 8 kWp peak power, which is a common size for an average German household [12]. The PV generator system is composed of the PV panels, maximum power point tracker (MPPT), and inverter that converts the generator's direct current (DC) power into AC power. The one-year data measured from a PV system installed in Munich, Germany is used as the PV generating profile. To implement the degradation of the PV system, a degradation factor of 0.5% of the PV's peak power per year is assumed [4,13].

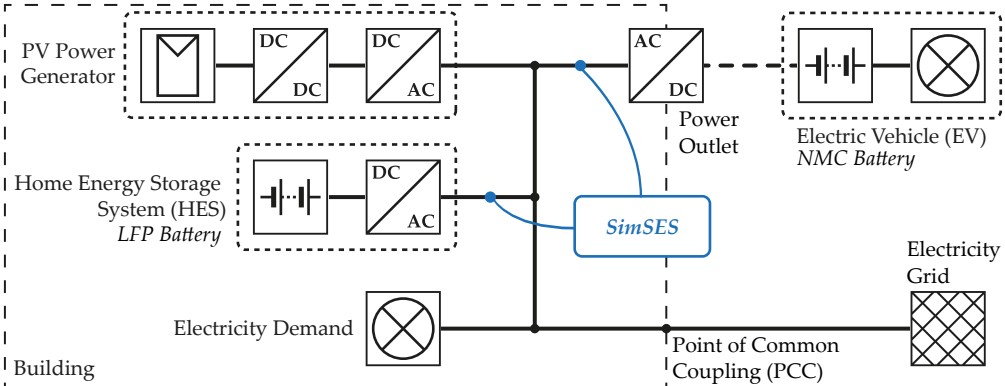

**Figure 3.** Residential power flow (RPF) model, consisting of the AC-coupled home energy storage system (HES), a photovoltaic (PV) power generator, electricity demand, the power outlet with the connected electric vehicle (EV), and the superordinate electricity grid. The simulation tool *SimSES* is used to validate the technical characteristics of the considered battery energy storage system (BESS).

In order to consider the electricity demand of a typical household, a representative one-year load profile (*profile* 31) out of a freely available set of smart-meter derived household load profiles is used in this study [14]. The annual electricity demand (only of the building, excluding that of the EV) of the considered household is set to 6000 kWh, a value taken from literature and well-suited to an average German household [12].

Further parameters and technical specifications of the household and its stationary HES can be taken from Table 1. The eligibility requirements, according to the German Federal Ministry of Economics and Technology, stipulate a feed-in limitation of 50% for PV generators that are operated in combination with a stationary or decentralized BESS [15]. Furthermore, a fixed feed-in remuneration price of 0.123 EUR/kWh is utilized, which is fixed and guaranteed for a period of twenty years [13]. Due to the projected electricity price of 0.437 EUR/kWh in 2030 and the electricity price of 0.294 EUR/kWh in 2018, a compound annual growth rate (CAGR) of 3.35% is assumed for the electricity purchase price in the simulation [16].

**Table 1.** Main parameters for the prosumer building and the home energy storage system (HES).

| Parameter | Value |
|---|---|
| Annual electricity demand | 6000 kWh [12] |
| PV peak power | 8 kWp [12] |
| Feed-in limitation | 50% [15] |
| Feed-in remuneration | 0.123 EUR/kWh [13] |
| Initial electricity price | 0.294 EUR/kWh [16] |
| Electricity price CAGR | 3.35% [16] |
| Battery chemistry | lithium iron phosphate (LFP) |
| Nominal energy content | 9 kWh [12,17] |
| SOC limitation | 5%, 95% [12,17] |

Lithium ion batteries (LIBs) are assumed for both the EV and HES. The cell chemistry chosen for the stationary HES within the building is based on a lithium iron phosphate (LFP) cathode and graphite anode. This chemistry allows a high cyclic stability [18], which makes it a suitable candidate for stationary applications [17].

The average German household with a HES has a usable energy content of 8.1 kWh [12]. From this the nominal energy content of 9 kWh is derived with the state of charge (SOC) limitations of 5% and 95% [17]. Furthermore, a self-discharge rate of 0.6% of the nominal energy content per month is assumed for the LFP cell [17]. Efficiency losses during charge and discharge processes of the battery are calculated via *SimSES*' equivalent circuit model, which depends on charging and discharging current, battery temperature, and SOC [10].

The semi-empirical degradation model of the LFP cell is also incorporated in *SimSES*. Degradation analysis is based on a superposition of calendar and cycling-related capacity fade [19]. During idle periods only calendar degradation, whereas during load periods also cyclic degradation is occurring [20]. This cyclic degradation is a function of multiple factors, including the depth of cycle (DOC), current, SOC range, and temperature [10]. A constant ambient temperature of 25 °C is assumed throughout the simulation period as the HES is installed within the building.

Since the AC coupling topology is the dominant topology for HESs in Germany [12], this setup is also used in this work. One of the major advantages of this topology over a DC coupling to the PV generator is an easy integration into a building with an existing PV generator, thus ensuring a high level of flexibility [21].

For the power-electronics efficiency, a simplified constant value of 95% is assumed in the RPF model. In order to make more accurate statements about the efficiency of the BESSs, the *SimSES* simulation tool takes into account a concave efficiency curve, which is derived from previous literature [4,22]. This curve considers the dependence on the inverter's output power and the fact that values below 10% of the rated inverter power result in a significantly lower efficiency.

Analogous to the procedure for the stationary HES, the power flows to and from the EV are optimized using the RPF model and then validated in *SimSES*. For all simulations of the EV and the ICE vehicle, a *B-segment* small car is considered [23–25]. An overview of the technical characteristics for the considered vehicles can be found in Table 2.

A nickel manganese cobalt (NMC) based cathode cell chemistry is chosen for the EV's BESS. Compared to other LIB cell chemistries, the NMC cell offers a higher energy density. The nominal and usable energy contents of the chosen EV battery, 21.6 kWh and 18.8 kWh, are closely linked to numbers often stated for EVs widely used in Germany. Derived from the nominal and usable energy contents, SOC boundaries of 8% and 95% are defined [17]. Similar to the LFP cell of the HES, the self-discharge rate of the NMC cell is set to 0.6% of the nominal energy content per month. Both the RPF model and detailed simulations using *SimSES* assume a round-trip efficiency of 95% for the EV battery [26].

In comparison to the highly sophisticated battery model of the LFP cell, the EV's battery is modelled using a more generic approach within *SimSES* [10]. Similar to previous work, a Wöhler curve (i.e., stress-number (S-N) curve) based fatigue model is used as the underlying method to estimate cycling-induced stress in the battery [4]. This method leads to an exponential weighting of DOC, i.e., an increased DOC leads to an overproportional increase in battery stress level, which again results in a reduced amount of equivalent full cycles (EFC) compared to low DOC values; thus, resulting in a shortened battery lifetime [27].

**Table 2.** Parameters for the electric vehicle (EV) and the internal combustion engine (ICE) vehicle.

| Parameter | Value |
|---|---|
| Vehicle class | B-segment small car [23–25] |
| Battery chemistry | nickel manganese cobalt (NMC) |
| Nominal energy content | 21.6 kWh [28] |
| Useable energy content | 18.8 kWh [28] |
| Battery round-trip efficiency | 95% [17] |
| Annually driven distance | 13,922 km [29] |
| Electricity consumption | 12.9 kWh/100 km [28] |
| Fuel consumption | 5.3 L/100 km [30] |
| Initial fuel price | 1.45 EUR/L [16] |
| Fuel price CAGR | 2.25% [16] |

The annual mileage of a passenger car is based on the German average, which is 13,922 km [29]. Therefore, a comparable EV, which consumes 12.9 kWh/100 km, requires approximately 1800 kWh annually [28]. In this paper, a gasoline-powered vehicle with an average fuel consumption of 5.3 L/100 km is used [30]. Analogous to the electricity costs, a temporally dynamic behavior is also assumed for the fuel price: An initial price of 1.45 EUR/L fuel is assumed for the start of the simulation. Due to the projected gasoline price of 1.89 EUR/L in 2030 and the gasoline price of 1.45 EUR/L in 2018, a CAGR of 2.25% is assumed for fuel prices in the simulation [16].

As part of this work, two EV profiles are created synthetically. The profiles for the two considered EVs (commuter and supplementary vehicle) are based on the US06 driving cycle and 83 charging profiles provided by the Forschungsstelle für Energiewirtschaft e. V., which are used in the federal study *Mobility in Germany* [9,31,32]. Both vehicle utilization patterns consist of a driving profile and a binary time series, which indicates whether the vehicle is connected to the power outlet of the building. It is assumed that the EV is only charged at the residential building and this additional electricity demand is directly allocated to the total electricity consumption of the household.

In Figure 4 an exemplary week (Monday to Sunday) in early summer is illustrated. The dashed areas in the two lower subplots show the plug-in times of the two utilization patterns, where the respective EV is connected to the building. As is immediately apparent, both profiles differ strongly in terms of their total plug-in time and respective daytime behavior: The commuter profile is only rarely connected to the building's power outlet during times of high solar irradiation on weekdays,

which makes it more difficult for this vehicle user to directly utilize surplus PV power. Instead, the cumulative plug-in time of the supplementary car is much higher, so the potential of optimizing the power flows between building and vehicle is assumed to be higher.

In order to bring the difference of the vehicle utilization types into a quantifiable context, the quotient between plug-in time and the residual power is formed. Residual power is defined as the difference between PV power and demanded power. For the two types of examined profiles, the resulting correlation coefficients are 7% for the commuter vehicle and 28% for the supplementary car. With the increased plug-in time, the BESS availability of the EV is increased, which increases the degree of freedom for power flow optimization. This increased utilization coefficient leads to a reduction in electricity purchases, which in turn lowers the OPEX of the prosumer. Based on this theory, this metric is introduced and discussed further in the following sections.

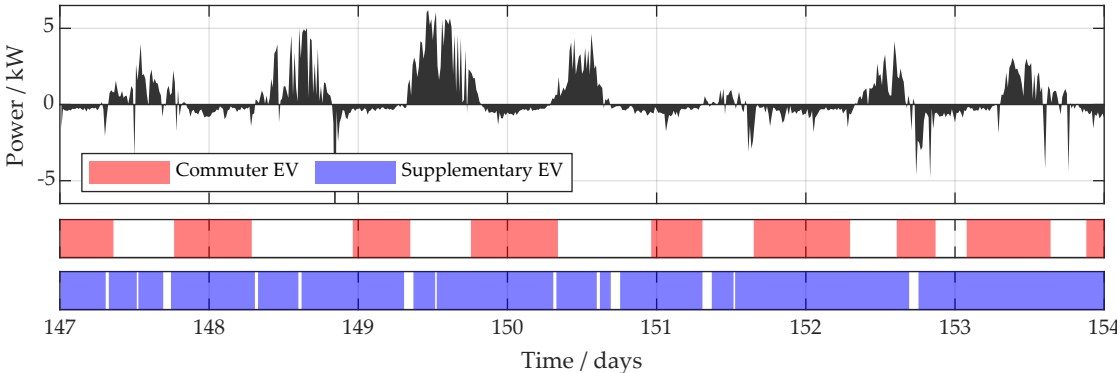

**Figure 4.** Residual power of exemplary week (Monday to Sunday) where photovoltaic (PV) excess power is characterized by positive values and the associated plug-in times (colored area) of the electric vehicles (commuter EV = red, supplementary EV = blue).

In addition to the two aforementioned vehicle utilization patterns, three different EV charging schemes are introduced. All three strategies are discussed in the context of storage system efficiency, degradation, and economic impact:

- Simple charging (SC): A simple rule-based charging of the EV is applied, where power is delivered unidirectionally from the power outlet of the building to the vehicle. As long as the vehicle is connected to the building and the EV's battery SOC has not reached the maximum SOC limit, the EV gets charged at the maximum allowed charge rate. The RPF model, as well as the simulation tool *SimSES*, are considering constraints for the respective SOC and C-Rate boundaries.
- Optimized charging (OC): Similar to SC the power outlet is used for unidirectional vehicle charging only. An advanced strategy is used that optimizes and controls the amount of energy and the timing of the EV's charging. The controller is fed by input values such as power flows within the building and the plug-in times of the EV.
- Vehicle-to-building (V2B): As an extension of the OC strategy, V2B enables a bidirectional power flow between the EV and building.

The RPF model's objective is to maximize the profit from the electricity sold and purchased throughout the simulation period. This comes down to a minimization of the OPEX of the prosumer. All scenarios use the following base objective function:

$$Max \sum_i \left( E_i^{\mathrm{r}} \cdot p_i^{\mathrm{r}} - E_i^{\mathrm{p}} \cdot p_i^{\mathrm{p}} \right) \tag{1}$$

whereby $E_i^{\mathrm{r}}$ denotes the amount of electricity that is sold to the superordinate electricity grid at time step $i$. The purchased electricity per time step is defined by the variable $E_i^{\mathrm{p}}$. The price signals $p_i^{\mathrm{r}}$ and $p_i^{\mathrm{p}}$ describe the remuneration and purchasing price at time $i$. Considering changing electricity prices

over time, price signals are time-dependent. Besides the objective function, inequality constraints for the BESSs' SOC and C-Rate, as well as equality constraints for the power flows at each node are considered and derived from a previous contribution [33].

Literature shows that the total cost of ownership (TCO) for an EV in Germany depends on many factors [25]. Due to the perennial lifetime of modern BESSs and the complex estimation of future BESS investment costs, capital expenditures (CAPEX) are neglected. In order to make the results as comprehensible as possible, only electricity costs and fuel costs are taken into account.

## 3. Results

The simulation results are presented and discussed in the following section. In total, fourteen different scenarios are conducted. As shown in Table 3, three different charging schemes, two vehicle usage patterns, and either one or two BESSs within the system are considered. The results are discussed in the context of battery degradation, storage system efficiency, and overall economic assessment, from the perspective of operating expenses for the prosumer.

**Table 3.** Overview of the fourteen simulated scenarios with three different charging schemes, two vehicle usage patterns, and either one or two BESSs within the prosumer household.

| Vehicle Usage Pattern | ICE | ICE w/HES | SC | OC | V2B | SC w/HES | OC w/HES | V2B w/HES |
|---|---|---|---|---|---|---|---|---|
| Commuter | yes (*ICE*) | yes (*ICE*) | yes | yes | yes | yes | yes | yes |
| Supplementary | | | yes | yes | yes | yes | yes | yes |

### 3.1. Economic Assessment of OPEX

As a first metric, the scenarios are evaluated and discussed from an economic perspective. Here, the OPEX for a short-term period of one year and a longer-term ten-year period are considered.

During the first year, even the EV scenario with the highest OPEX, the SC scheme, showed a cost reduction of 31% without HES compared to the ICE vehicle without HES. With the addition of a home energy storage system to the scenarios, the OPEX reduction when using the SC scheme is 39% (EUR 571) in comparison to the ICE vehicle with the same HES.

As illustrated in Figure 5, strong differences between EV charging strategies can be detected. Both without and with HES, the implementation of an optimized charging (OC) scheme leads to a reduction in OPEX. Further cost improvements can be gained by allowing bidirectional power flows (V2B) between the building and the EV. This impact of optimized charging schemes (unidirectional and bidirectional) is particularly strong if there is no additional HES, leading to cost reductions of 14% and 23% in comparison to the SC strategy. The same ratios, with the addition of a HES, are reduced to 12% and 13% respectively.

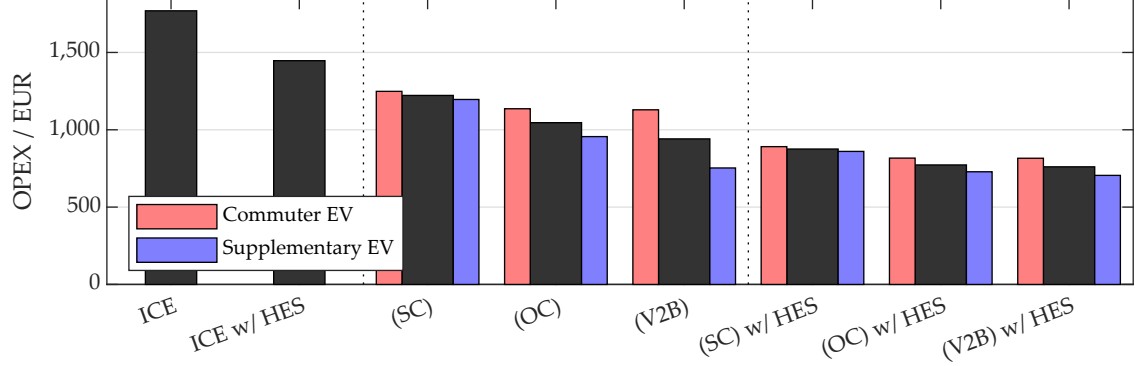

**Figure 5.** Operating expenses (OPEX) for one year. The dark-grey column represents the average value.

On average, OPEX decrease by 25% if, in addition to an EV, a stationary HES is available, resulting in EUR 115 cost reduction for the observed setting and year. Furthermore, the results for the commuter

and supplementary car in the V2B scenario without HES showed a strong difference. Due to the relatively higher plug-in time of the supplementary car (especially during periods of high PV power), more self-generated energy can be stored in the vehicle, which results in higher self-consumption and self-sufficiency rates that are illustrated in Figure 6. Additionally, the scenarios of the supplementary car, with or without an additional HES, result in almost the same costs. Again, the supplementary car's high amount of plug-in time increases the utilization of the vehicle battery, thus making the stationary HES almost obsolete.

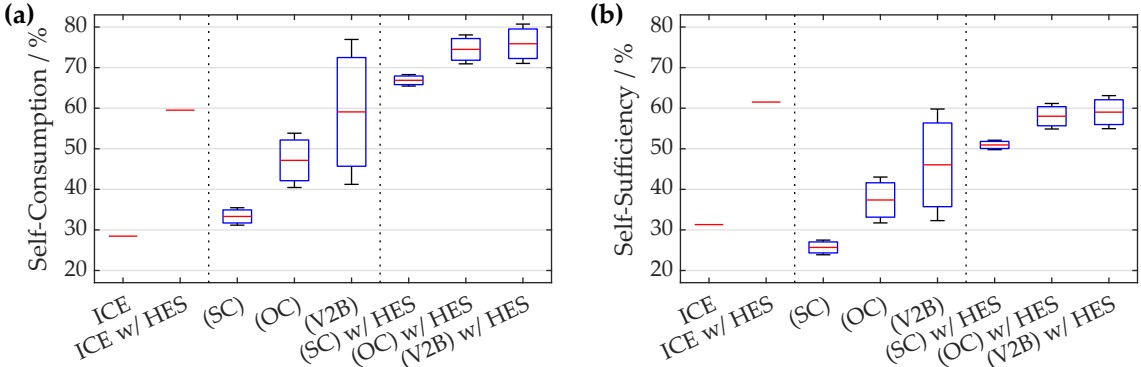

**Figure 6.** (**a**) Self-consumption and (**b**) self-sufficiency rate for the investigated scenarios. For both metrics, the top edge of each boxplot represents the supplementary car. The lower values of the boxplots are defined by the commuter car, which has a shorter plug-in time compared to the supplementary car.

As shown in Figure 7, the relative differences between the six EV scenarios remain almost the same as in the one-year view. The slight differences are due to the CAGR effect of rising electricity prices. However, the OPEX relationships between the ICE vehicle and EV changed because the expected fuel price increase is lower than that of electricity. A more detailed picture of the OPEX and their seasonal development over ten years can be seen in Figure A1 in the Appendix A.

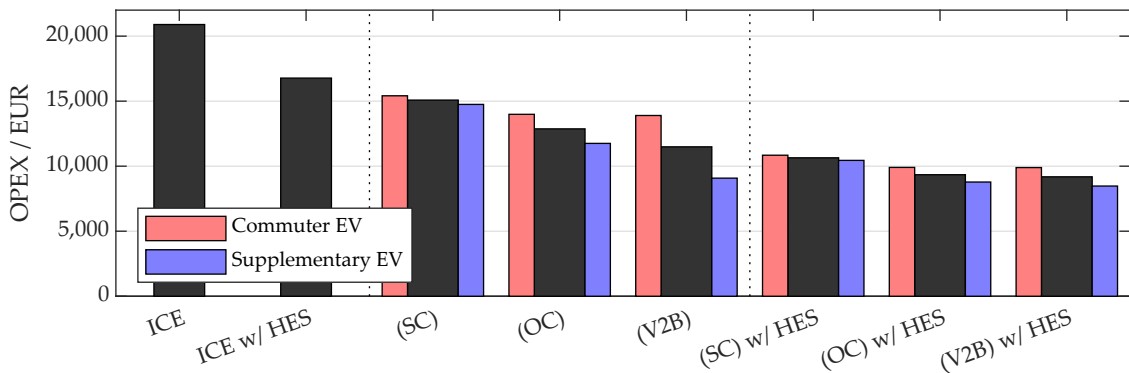

**Figure 7.** Operating expenses (OPEX) for ten years. Compound annual growth rate (CAGR) of energy costs are considered, so that costs for ten years are more than ten times the one-year costs. The dark-grey column represents the average value.

### 3.2. Battery Lifetime and Degradation

A common procedure when determining the end of life (EOL) of BESSs is reaching a certain capacity value. Specifically, values between 70% and 80% of the nominal battery capacity are often used to describe the EOL of the BESS [34,35]. In this work, the threshold of 80% is defined as EOL criteria, for both the HES and the EV battery. Figure 8 shows the battery degradation for both BESSs and the simulated scenarios. A more detailed evaluation of the degradation of the two battery types is discussed in the following paragraphs.

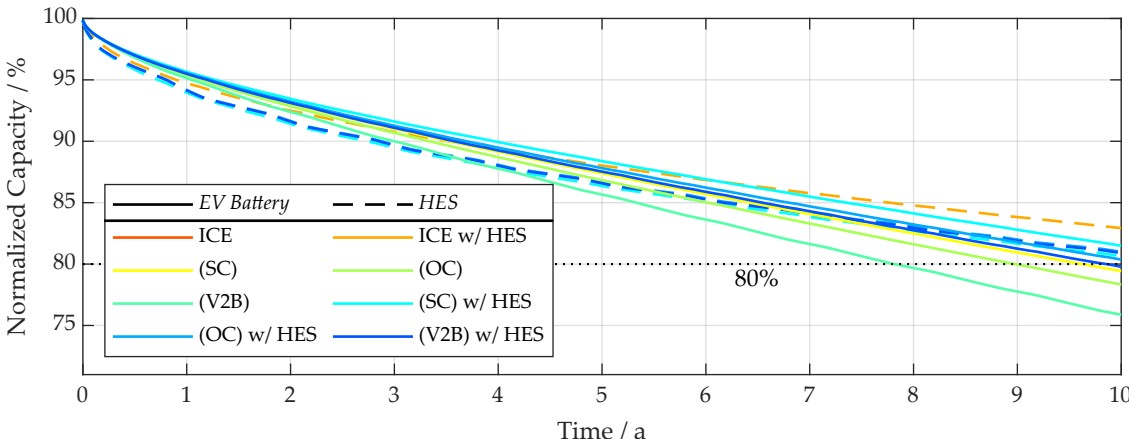

**Figure 8.** Remaining capacity of electric vehicle's battery (nickel manganese cobalt (NMC) cell chemistry, solid line) and home energy storage system (HES) (lithium iron phosphate (LFP) cell chemistry, dashed line) over ten years, with the highlighted end of life (EOL) threshold at 80% nominal battery capacity.

### 3.2.1. Home Energy Storage System

As visualized in Figure 9a the results of the observed scenarios show a lifetime between 10.7 years and 13.6 years for the battery of the HES. It is noticeable that the highest lifetime is achieved in the scenario of the ICE vehicle combined with a HES. For the EV scenarios, the lifetime is reduced by about 20%, whereby the simple charging scheme shows the shortest lifespan of 10.7 years. A further trend that can be seen in all three EV scenarios is that the battery lifetime in the scenarios with the supplementary car is always higher than the ones of the commuter vehicle. In both the OC and V2B strategy, this results in a relative lifetime improvement of about 6% for the supplementary car.

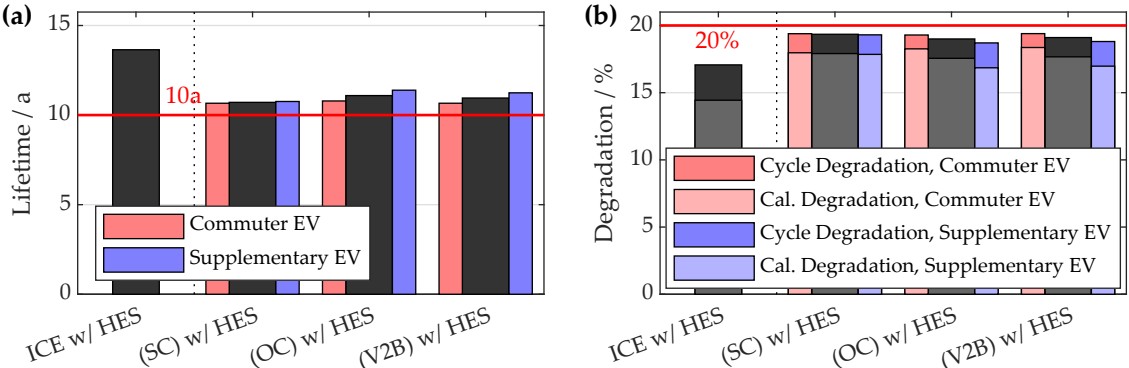

**Figure 9.** (**a**) Modelled lifetime of the lithium iron phosphate (LFP) home energy storage system (HES) with a nominal capacity of 9 kWh and (**b**) calendar and cyclic degradation during ten years of operation, with the end of life (EOL) condition of 80% remaining capacity. The dark-grey column represents the average value.

In Figure 9b, the relative calendar and cyclic degradation over the course of ten years of operation is illustrated. The results show that the 20% capacity fade is almost reached after ten years for the HES. In Figure 9a, it can be observed that a total lifetime of up to 13.6 years is reached. This can be explained by the initial intensity of degradation processes at the early stage of the battery's operation, which then decrease over time.

The fact that cells suffer particularly from SOC values in the lower and upper SOC range is reflected in the LFP model used for the simulations of this study [20]. Due to increased stress characteristics at these more extreme SOC regions, calendar degradation is accelerated. This, in turn

leads to a reduced lifetime. At roughly 90%, calendar degradation processes are the main driver for the reduced battery lifetime. On the other hand, the cyclic degradation stress is fostered by high amounts of EFC. It should also be emphasized that the measured values shown are not the only drivers for battery degradation.

The battery's EFC are especially significant for cyclic degradation. The four HESs of the observed scenarios show annual EFC values of between 167 to 246, as shown in Figure 10a. Especially in the SC scheme, the EFC are significantly higher than those of the other scenarios. The lowest and almost equal amount of EFC is achieved in the settings of unidirectional (OC) and bidirectional (V2B) optimized charging.

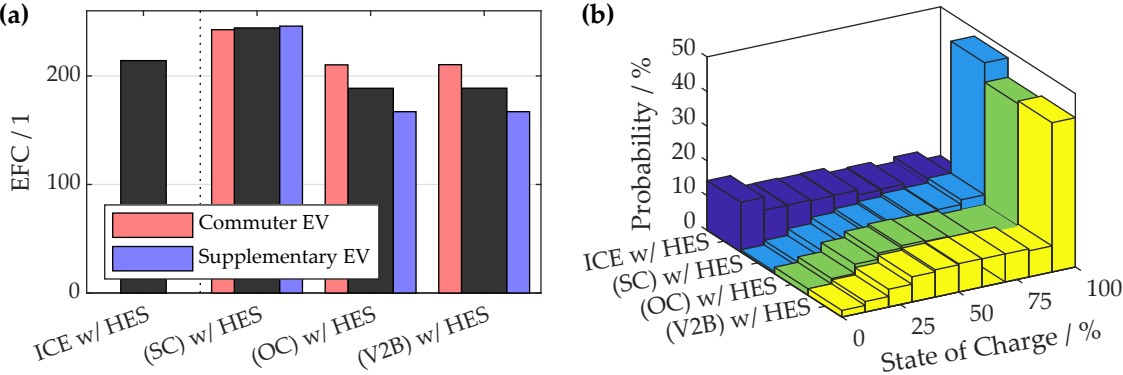

**Figure 10.** (**a**) Average amount of annual equivalent full cycles (EFC) of the home energy storage system and (**b**) probability distribution of the average state of charge (SOC) per scenario. The dark-grey column represents the average value.

Another degradation factor that is of importance for the lifetime of a LIB is the average SOC. This measure is illustrated in Figure 10b and gives insight into the probability distribution of the SOC for the four considered HESs. Here, a distinctive difference between the ICE vehicle and EV scenarios can be seen. While the SOC values of the HES have a rather homogeneous distribution in the ICE scenario, values in EV settings are much more heterogeneous. In all considered scenarios in which an EV and a HES are combined, it is shown that the SOC of the HES has a high probability density at high values. In the case of the simple charging (SC) scheme, the trend towards high SOC values is particularly strong. As with the number of EFC, here too, both scenarios OC and V2B show approximately the same, and better, results.

### 3.2.2. EV Battery

Like the evaluation of the HES's data, the battery of the EV is also examined with regard to degradation for the different scenarios. *SimSES* is used to model an isolated storage system behavior of the EV battery. Since the battery model used for the NMC cells is a generic model in comparison to the semi-empirical degradation model used for the LFP cells, results are shown in less detail for the EV battery.

A common standard for the expected lifetime and warranty period for EV batteries is seven to ten years [36]. Within this period, the remaining battery capacity should not fall below the defined EOL criteria of the battery. For the considered scenarios, it is shown that the EV battery has a lifetime of between 7.2 years and 11.8 years, as can be seen in Figure 11.

It is noticeable that its lifetime can be increased by an average of 19% if the EV battery works in conjunction with the stationary HES. The existence of a second storage system leads to a segmentation of the power flows, which results in a reduced stress level of the EV battery.

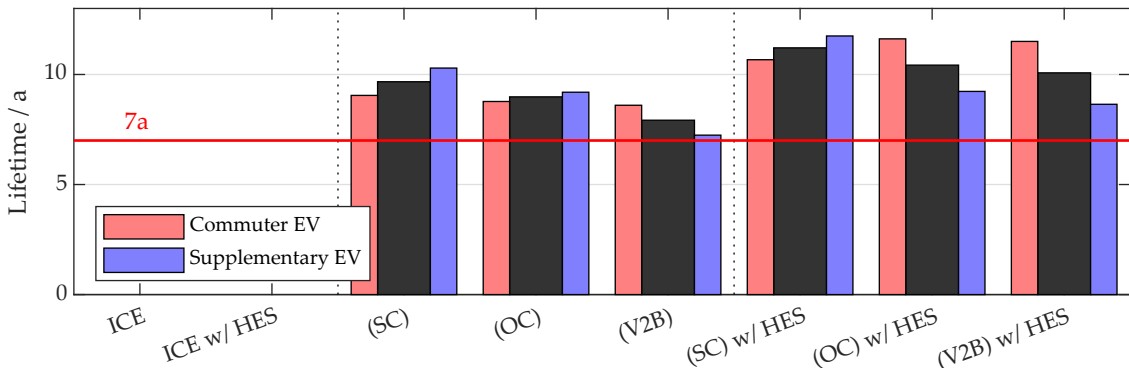

**Figure 11.** Lifetime calculation of the electric vehicle's nickel manganese cobalt (NMC) battery, based on a generic battery model, with the defined end of life (EOL) condition of 80% of the nominal capacity. The dark-grey column represents the average value.

The scenarios with SC and OC schemes show the same amount of EFC, due to the fact that in these unidirectional scenarios, only the power needed at a later time for driving is delivered from the building to the vehicle. Despite same amount of EFC of the EV battery in the SC and OC scenarios, the lifetime of the optimized charging (OC) scheme is reduced by 7%. For better interpretation along with the degradation model used herein (based on Wöhler curves), the average absolute values of the DOC are shown in Table 4. Here, it can be seen that the average DOC in scenarios with a HES decreases by about 30% compared to the same settings without a HES.

**Table 4.** Annual amount of equivalent full cycles (EFC) and the absolute depth of cycle (DOC) (normalized to the amount of EFC) of the battery taken as an average from the commuter and supplementary electric vehicle (EV).

|  | ICE | ICE w/HES | SC | OC | V2B | SC w/HES | OC w/HES | V2B w/HES |
|---|---|---|---|---|---|---|---|---|
| **EFC** | n/a | n/a | 85.5 | 85.5 | 119.3 | 85.5 | 85.5 | 89.5 |
| **\|DOC\|** | n/a | n/a | 1.00 | 0.98 | 0.98 | 0.58 | 0.76 | 0.76 |

The degradation in the case of V2B is significantly higher. Results show that the annual number of EFC at 119.3 increase by 40% when there is no additional BESS in the system besides the EV battery. This increase in EFC and the relatively high average DOC values result in a lifetime reduction of about 12% compared to the OC scheme.

For scenarios considering two BESSs, the V2B scenario again shows the highest battery degradation. Because of the permanently available HES, surplus PV power can also be stored in the stationary HES and therefore the number of EFC in the V2B scenario is only slightly higher than that of the unidirectional scenarios (SC and OC). However, the battery lifetime in the V2B case is shortened by about 3% compared to the same setting with OC scheme.

The commuter car battery in the V2B scenarios has a lower energy throughput and thus a lower number of EFC. The relatively higher plug-in time of the supplementary car allows more surplus energy to be charged into and discharged from the EV battery, resulting in a higher number of EFC and a reduced lifetime.

### 3.3. Storage System Efficiency

In addition to battery degradation, BESSs' round-trip efficiency values are also considered. For both BESS types, the stationary HES and the storage system of the EV, a round-trip efficiency of about 88% is achieved for all operational modes.

More detailed analysis reveals that the dominant source of storage losses comes from power-electronics. This is in line with efficiency analysis conducted on stationary storage systems [37].

Overall, between 8% and 10% efficiency losses are caused by the inverter. This emphasizes the relevance for optimizing the specifications of the technical components of a storage system.

Furthermore, storage losses are considered during the charging and discharging processes of the battery. Storage losses within the battery cells range from 2% to 4% in the considered simulations, which is in line with results from literature [17]. Self-discharge losses, which account for below 0.1% of the total energy throughput, play a subordinate role. This low percentage of storage losses is similar for both storage technologies in all scenarios.

## 4. Discussion and Conclusions

The following section summarizes the results derived from the simulations and discusses them in the context of previous literature. At the end of the section, related and future research fields are highlighted.

### 4.1. EV Versus ICE Vehicle

In the previously discussed results section it is shown that an EV can have a significant economic advantage compared to ICE powered vehicles when it comes to reducing electricity costs of a prosumer household. Considering a time span of ten years, it is shown that OPEX can be reduced by an average of 37% (without an additional stationary HES) and 42% (with HES). Even the least economically lucrative scenario with simple charging (SC) shows an average savings potential of 28% (without HES) and 37% (with HES) compared to the same scenarios with an ICE powered vehicle.

Looking at the average results of the individual EV scenarios, it can be said that the considered additional energy costs for the investigated ICE vehicle are about EUR 7400 higher than for its electric-powered counterpart, which may justify an investment in a higher priced EV. Of course, further cost components and economic and policy aspects must be taken into account in order to carry out a complete economic analysis [25,38]. Furthermore, at the moment, there is no consensus on when an EV is equivalent to an ICE vehicle in terms of investment costs.

In the context of battery lifetime, the simulations reveal a trend of stronger degradation when an EV is included in the consideration. The HES's battery reaches the defined EOL criterion earlier by 20%, on average, when an EV is connected to the household. Minimizing OPEX means that more self-generated energy is stored in the HES. In the EV scenarios, the effect leads to an increased occurrence of high SOC levels, which accelerates internal degradation processes of the LFP cells [10]. In order to compensate this effect, the developed charging strategies must be further optimized.

Furthermore, the share of automotive batteries that are used for further applications after their primary use as an EV battery is growing. Particularly, the installation and operation of such second-use batteries in stationary applications is increasing [39]. This use of second-use batteries allows an additional economic impact of the BESS, which makes it more lucrative for their stakeholders [40].

### 4.2. Impact of Vehicle Utilization Pattern

From the simulations it can be concluded that the supplementary vehicle type has a beneficial effect on electricity cost reductions. This is shown by the lower OPEX in all scenarios when compared to the commuter EV, which has less plug-in time at the building. This relation confirms the initial theory that a higher correlation coefficient between residual power and plug-in time leads to an economic improvement. It is expected that, from the perspective of an office building with PV generation, the connected EVs from commuting employees would have the same beneficial outcome. The underlying effect can also be explained by the household's increased self-consumption and self-sufficiency rate with the supplementary vehicle profile [7]. On average, OPEX in the commuter car scenarios are about 16% higher than those with the supplementary vehicle. This cost increase is particularly high when considering a bidirectional charging scheme (V2B).

In terms of battery degradation, on the other hand, it is shown that battery lifetime of the HES is slightly increased in the commuter car scenarios. However, the average battery lifetime for the

EV battery shows a favorable behavior in the supplementary scenarios, in particular during V2B charging schemes. DOC values and the underlying Wöhler curve for the EV battery degradation model represent the main drivers for this effect [27].

### 4.3. Impact of Considering an Additional HES

Previous literature has shown that it is still difficult to operate a HES in Germany in an economically lucrative way [4]. Although the results presented in this paper only relate to OPEX, it is noteworthy that a HES can reduce these costs by an average of 23% during the first year. When taking into account the rising electricity retail tariff estimated for the next ten years [16], the cost savings may rise by another few percentage points.

Due to the segmentation of power flows when considering a HES, both the energy throughput and relative DOC values of the EV battery can be reduced. The reduced stress level leads to an increase of the EV's battery lifetime by an average of 20%.

Whether and to what extent the advantages of the lifetime extension of the EV battery and OPEX reduction justify additional expenses of a HES depend, in turn, on the CAPEX. Taking into account the discussed prosumer and an operation period of ten years, HES investment costs below EUR 2305 (V2B scenario) and EUR 4437 (SC scenario) would be justified. The higher value in the SC scenario results from the fact that, here, an additional HES has a higher potential for OPEX improvement, which is further discussed in the subsequent paragraphs. Assuming steadily declining CAPEX for stationary battery packs [41], a HES can become increasingly interesting for residential buildings. If, in addition to the minimization of OPEX and self-consumption improvements, other applications are served, the economics of the HES can be increased even further [42].

### 4.4. Impact of Charging Scheme

Both in the scenarios with and without HES, the simple charging (SC) scheme resulted in the highest OPEX. The condition that the EV is charged as soon as it is connected to the building also results in overall low self-consumption and self-sufficiency rates of 33% and 26%, which are illustrated in Figure 6.

In the optimized charging (OC) scheme with a HES, the electricity costs can be reduced by 12% (about EUR 1300 for a ten-year operation period). This effect is even more pronounced when there is no additional HES and the EV battery is the only BESS in the setting. OPEX can be reduced by 15% (about EUR 2200) compared to the SC scheme when the EV battery is the only storage unit to decouple energy supply and demand.

By allowing a bidirectional power flow between the building and the EV (V2B) instead of the unidirectional power flow (OC), further cost savings can be achieved. Relative to the OC scheme, this results in a further OPEX reduction of 2% (with HES) and 11% (without HES). Analogous to the above comparison between the SC and OC schemes, there is an increased cost saving potential if the EV battery is the only storage unit in the system. When considering the absolute values of the savings potential, an OPEX reduction of EUR 160 (with HES) and EUR 1381 (without HES) results for an operation period of ten years. This comes at the cost for additional upfront investment costs: The low savings potential of the scenario with HES suggests that the additional investment costs for a power outlet with bidirectional power flow are difficult to compensate. On the other hand, in scenarios with a single EV battery, the V2B scheme could be economically lucrative in comparison to the OC scheme, if the additional investment costs are below the cost savings of EUR 1381.

In contrast to improved electricity expenditures, the lifetime of the EV battery decreases in the OC and V2B schemes. Due to increased energy throughput, particularly in the V2B scheme, the lifetime is reduced by up to 12% compared to the optimized unidirectional charging (OC). SC scenarios lead to the highest lifetime with a relative improvement of 7% compared to the OC scheme. One of the main drivers for the increased degradation are the relatively higher DOC values in the OC and V2B scheme. In addition, two more obstacles come into play: The prediction of power values is needed for

effective OC and V2B schemes. Furthermore, automotive original equipment manufacturers (OEMs) provide warranties on the use of the EV battery for vehicle purposes. While this is maintained in the SC and OC schemes, the EV battery in the V2B scheme does not only function as an EV battery, but also as a buffer storage unit for the whole prosumer household. Thus this could pose a challenge to incentivizing V2B schemes.

The EV market shows a trend towards increasing battery capacity. EVs being manufactured currently are often more than twice as large in terms of nominal energy content than the 21.6 kWh EV battery that is considered in this work. It can be expected that the higher cost savings and the lower necessity of an additional HES due to the V2B scheme will be enhanced with these increased capacities.

### 4.5. Limitations and Future Research

The discussed simulations are conducted assuming perfect foresight of energy supply and demand, both for the household and the vehicle. In order to emphasize on limited foresight, other algorithms can be used. For instance, in [43] a fuzzy logic controller (FLC) is presented for a V2B environment.

Since the discussed RPF model is using constant efficiency values, it would be an improvement to implement non-linear relationships, as already implemented in *SimSES* [10]. This step would improve the RPF's validity, but would also lead to an increasing complexity of the optimization algorithm, resulting in an elevated computation time.

Although the generic battery model of the NMC cell provides values for the battery degradation [4,27], the quality of the battery model can be improved further. In comparison to the current model, more sophisticated degradation models could be implemented, as done for the semi-empirical degradation model of the LFP cell [20].

In order to generate a more profound insight into the economic results of the discussed settings, further research should take additional cost components into account. Although it is not clear how the CAPEX for batteries will develop in the future, there are estimations in recent literature that could be used [41]. Another cost component that is of relevance in that perspective are battery degradation costs [44]. The consideration of these factors would provide a more complete picture of the total cost of ownership (TCO), which in turn would allow for more precise conclusions.

**Author Contributions:** S.E. developed the conceptualization, including the generation of the optimization framework for the residential power-flow model, coordinated the study, and wrote the manuscript. H.H. significantly contributed to the conceptualization and methodology of the study. D.K. offered support in using simulation tool SimSES. A.J. provided overall guidance for the study and contributed to fruitful discussions on the methodology. All authors have read and approved the manuscript.

**Funding:** This research is funded by the Bavarian Ministry of Economic Affairs, Energy, and Technology via the research project StorageLink (grant number IUK-1711-0035//IUK551/002), and the Technical University of Munich (TUM) in the framework of the Open Access Publishing Program.

**Acknowledgments:** The authors thank Sebastian Fischhaber from the Forschungsstelle für Energiewirtschaft e. V. for providing electric-vehicle charging profiles.

**Conflicts of Interest:** The authors declare no conflict of interest. The funders had no role in the design of the study; in the collection, analyses, or interpretation of data; in the writing of the manuscript; or in the decision to publish the results.

## Abbreviations

The following abbreviations are used in this manuscript:

| | |
|---|---|
| AC | Alternating current |
| BESS | Battery energy storage system |
| CAGR | Compound annual growth rate |
| CAPEX | Capital expenditures |
| DC | Direct current |
| DOC | Depth of cycle |
| EFC | Equivalent full cycles |

| | |
|---|---|
| EOL | End of life |
| EV | Electric vehicle |
| HES | Home energy storage system |
| ICE | Internal combustion engine |
| LFP | Lithium iron phosphate |
| LIB | Lithium ion battery |
| LP | Linear programming |
| NMC | Nickel manganese cobalt |
| OC | Optimized charging |
| OPEX | Operating expenses |
| PV | Photovoltaic |
| RPF | Residential power flow |
| SC | Simple charging |
| SOC | State of charge |
| V2B | Vehicle-to-building |

## Appendix A

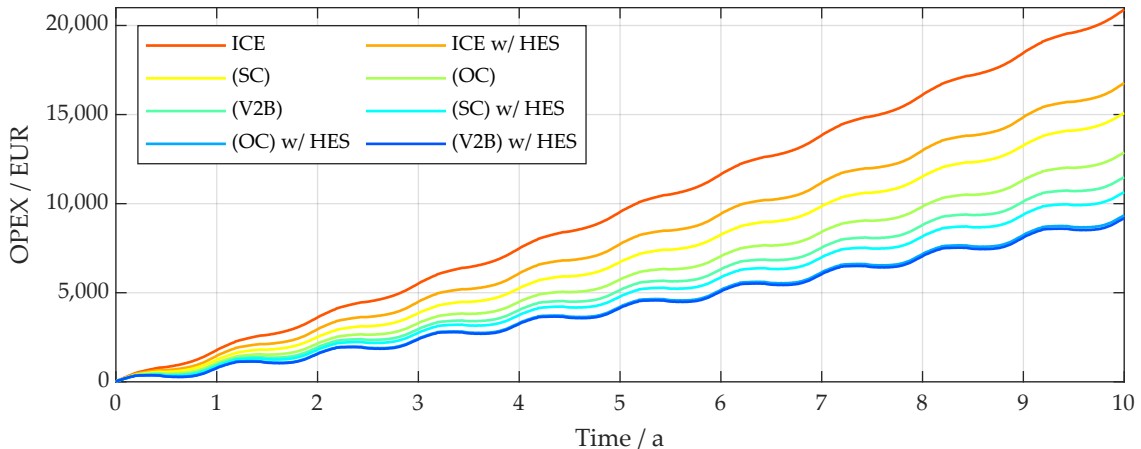

**Figure A1.** Operating expenses (OPEX) during ten years of operation showing a strong seasonal pattern.

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
