# Peer review of "A Techno-Economic Analysis of Vehicle-to-Building: Battery Degradation and Efficiency Analysis in the Context of Coordinated Electric Vehicle Charging"

_energies, doi:10.3390/en12050955_

Round 1

Reviewer 1 Report

The authors presents an interesting techno-economic study of the energy storage system in low voltage application. The paper is well written and the research activity is very relevant and actual.

The paper conducts the research based on a tool SimSES for different scenarios .

In my opinion, this is an interesting paper, however the pertinence of the results depend strongly of the accuracy of the battery degradation models used in the tool, This issue should be clarified.

How the model  parameters of the battery were determined. Which is the impact of the level of current used in charge and discharged cycles. I think that is fair to the reader to know the assumptions used in the tool and conditions.

In the methodology sections, it is unknown how the coordinated electric vehicle charging is implemented. This aspect must be clarified.

I would like to suggest to authors to consider in the study he scenario of using at home storage system based on battery second life.

Author Response

The authors thank the editors and experts who reviewed the paper for their valuable comments, which we have taken into account to produce an improved version of the manuscript. In order to answer the reviewers’ comments in a tractable way, these are answered one-by-one in the following paragraphs.

Reviewer 2 Report

The paper solves an interesting problem: battery degradation and efficiency analysis of the V2B technique. It is well conducted and well presented, with new contributions to the relevant literature. I think this paper can be published in its current form or minor editorial changes (does not need to be reviewed again).

 ****************

 The x-axis of Fig. 8 is [Time/a]. Please double check.

 Given that accurate aging models are usually difficult to obtain, it would be interesting to discuss how to use the proposed method in the abscence of an accurate degradation model.

Author Response

(The authors gave the same response as above.)
